# A protocol for a systematic review and meta-analysis of the effect of muscle energy techniques on shoulder joint pain

Ye Ji Kim[1]*, Seojae Jeon[2]*, Hyeonjun Woo[3], Won-Bae Ha[4], Junghan Lee[4]

1 Chuna Manual Medicine Research Group, College of Korean Medicine, Won-Kwang University, Iksan-si, Jeonbuk-do, Republic of Korea, 2 Korea Institute of Integrated Medical Research, Jangheung-gun, Jeollanam-do, Republic of Korea, 3 Department of Korean Medicine Rehabilitation, College of Korean Medicine, Semyung University, Chungju-si, Chungcheongbuk-do, Republic of Korea, 4 Department of Korean Medicine Rehabilitation, College of Korean Medicine, Wonkwang University, Iksan-si, Jeonbuk-do, Republic of Korea

☯ These author contributed equally to this work.
* fr1771@naver.com (SJ); max2869@naver.com (YJK)

## Abstract

### Introduction

Muscle energy techniques (MET) for shoulder muscles improve both shoulder muscle tension and the range of motion of the glenohumeral joint. This systematic literature review will investigate the effects of MET on shoulder pain as a result of muscle tension in the glenohumeral joints of patients with shoulder disorders and collect clinical evidence regarding the effectiveness of muscle energy techniques on glenohumeral joint pain. Based on previous studies, we anticipate that MET may significantly affect shoulder joint pain. We expect to provide moderate to high levels of evidence regarding the effectiveness of MET in the treatment of shoulder pain.

### Methods

Nine electronic databases will be searched for articles published up to November 2024, including PubMed, EMBASE, CENTRAL, KCI, KISS, KMbase, RISS, DBpia, and OASIS. Search terms will consist of terms related to the outcome (e.g., "shoulder") and intervention (e.g., "muscle energy technique," "post-isometric relaxation," "isometric stretching"). Studies selected for the systematic review and meta-analysis will include randomized controlled clinical trials and studies using MET applied to the human shoulder muscles. Qualitative and case studies will be excluded. Two authors will independently assess each study for eligibility and risk of bias and extract the data. This study will analyze the effects of MET on shoulder pain. Additionally, we intend to demonstrate the effect size of muscle energy techniques on factors such as range of motion. Our study will provide clinical evidence for the effects of muscle energy techniques on shoulder joint pain. Our study aims to provide clinical evidence supporting the moderate-to-high effectiveness of MET in treating shoulder joint pain.

**Data availability statement:** Deidentified research data will be made publicly available when the study is completed and published.

**Funding:** This research was supported by a grant of the Korea Health Technology R&D Project through the Korea Health Industry Development Institute (KHIDI), funded by the Ministry of Health & Welfare, Republic of Korea (grant number: RS-2023-KH142004).

**Competing interests:** The authors have declared that no competing interests exist.

## Prospero registration

ClinicalTrials.gov CRD42024532367

## Introduction

The shoulder complex consists of four joints, the acromioclavicular, sternoclavicular, glenohumeral, and scapulothoracic joints, formed by articulation of the clavicle, scapula, ribs, humerus, and surrounding soft tissues such as muscles and ligaments. These structures produce highly coordinated movement across multiple joints. The shoulder complex prioritizes mobility over stability and offers the greatest range of motion (ROM) among the joints of the human body. Consequently, due to its anatomical instability, it is prone to pain and dysfunction [1].

The disruption or weakness of any muscle in this region can impede the inherent continuum of motion, resulting in pain [2]. Therefore, when considering shoulder pathology from a functional perspective, the entire shoulder complex must be investigated to restore muscle balance. Currently, surgical interventions dominate the treatment of shoulder disorders. However, these surgeries often lead to secondary complications, including stiffness and arthrofibrosis [3,4].

One method for minimizing surgical side effects, alleviating pain, and enhancing functional movement is the muscle energy technique (MET) [5,6]. The MET aims to reduce pain and improve restricted joint mobility [7]. Its mechanism involves post-isometric relaxation and reciprocal inhibition following muscle contraction. Muscle and joint tension decrease within approximately 15 s of isometric contraction, thus facilitating the natural expansion of joint movement [8]. Additionally, muscles treated with MET exhibit increased elasticity, whereas the joint capsule and surrounding tissue may elongate. Consequently, the elasticity and structure of the muscles and tissues change, ultimately increasing the joint range of motion and alleviating shoulder joint pain. Furthermore, MET-induced joint movements enhance proprioceptive feedback and improve motor control and learning abilities [9].

Because of these advantages, the MET is widely used in clinical settings, particularly in patients with adhesive capsulitis and upper crossed syndrome. Randomized controlled trials (RCTs) have been conducted on MET [10,11]. However, studies analyzing the effects of the shoulder joint MET on pain and ROM in all patients are lacking. Although numerous studies have analyzed the effects of massage, shoulder joint mobilization, manual therapy such as the Kaltenborn mobilization and Mulligan techniques, and stretching exercises on shoulder ROM and pain, no independent study has analyzed the effects of MET alone on shoulder joint pain and ROM [12–15]. Therefore, this study aims to analyze the effect size of MET techniques on shoulder joint pain and function through a systematic literature review and meta-analysis to comprehensively consolidate previous research findings.

## Methods

### Registration of this study

The proposed systematic review was formally registered with the International Prospective Register of Systematic Reviews (PROSPERO) under the registration number CRD42024532367 on May 5, 2024. This study will involve a systematic review and update according to this protocol. This protocol strictly adheres to the guidelines outlined in the Preferred Reporting Items for Systematic Review and Meta-Analysis Protocols 2015 statement [16] as well as the Cochrane Handbook for Systematic Reviews of Interventions [17]. Any

modifications to previously published protocols will be explicitly noted, accompanied by a thorough delineation of the amendments.

## Data sources

Two independent researchers (YJK and SJJ) will comprehensively search nine databases from their inception up to November 2024. The study will encompass three English-language databases: MEDLINE via PubMed, EMBASE via Elsevier, and the Cochrane Central Register of Controlled Trials CENTRAL), and six Korean-language databases: Korea Citation Index (KCI), Korean Studies Information Service System (KISS), Korean Medical Database (KMbase), Research Information Service System (RISS), Data Base Periodical Information Academic (DBpia), and Oriental Medicine Advanced Searching Integrated System (OASIS). In addition, we will explore the reference lists of pertinent articles and conduct manual inquiries using Google Scholar to identify further contributions. Our search will encompass both peer-reviewed journal literature and "gray literature" such as theses and conference proceedings.

## Search strategies

The search terms will comprise the disease term (e.g., "shoulder impingement syndrome" and "rotator cuff tear") and intervention term (e.g., "muscle energy technique on shoulder"). Table 1 shows the search strategies for the PubMed and EMBASE databases, which will be modified and used similarly for other databases.

## Inclusion criteria

RCTs will be included in this systematic review and meta-analysis. Qualitative studies and case studies will be excluded from the analysis.

We will include studies of patients with shoulder pain attributed to shoulder joint disorders without restrictions on sex, age, or race. Studies using METs applied to human shoulder muscles will be included with no restrictions on comparator conditions (no treatment, stretching, or other manipulations).

**Table 1. Search strategy according to PICO.**

| Criteria | Standard contents |
|---|---|
| Research method | Randomized controlled clinical trials |
| Research design | Randomized controlled clinical trials |
| Purpose | Research purposes will be revealed. |
| Participants/Patients | Patients with shoulder movement problems. No restriction will be placed on the sex, race, or age of the participants. |
| Intervention/Moderate variables | Muscle energy technique applied to the human shoulder muscles |
| Comparison | Placebo and blank control |
| Outcomes | -Primary Outcomes<br>Visual Analogue Scale |
| | -Secondary Outcomes<br>Range of motion of shoulder joint<br>Shoulder pain and disability index<br>Quality of life |
| Data statistics | All figures depicting mean, standard deviation, and t and f values, to calculate effect size |

## Types of outcome measures

**Primary outcome.** The primary outcome measure will be the visual analog scale (VAS) score developed by Hayes and Patterson in 1921, which is used to subjectively evaluate acute and chronic pain, similar to other assessment tools used to evaluate shoulder joint pain in each study. Scores are documented by marking a 10-cm line, symbolizing a spectrum from "no pain" to "worst pain," through handwritten notations adhering to academic conventions [18].

**Secondary outcome.** Tools for evaluating range of motion (ROM) and function related to the shoulder joint will be employed to assess the secondary outcomes.

1) The ROM of the shoulder joint measured using a universal goniometer will be included as a secondary outcome, focusing on the effect on daily activities, including flexion, extension, and abduction of the shoulder joint [19].

2) Roach et al. developed the shoulder pain and disability index (SPADI), a self-administered questionnaire consisting of 13 items, to measure shoulder pain and disability [20]. The questions are classified into two subscales: a 5-item subscale that measures pain and an 8-item subscale that measures disability. The SPADI has two versions: one is scored on a VAS and the other is scored on a numerical rating scale (NRS). Each subscale is totaled and converted to a score of 100. The average of the two subscales is calculated to obtain a total score of 100, with higher scores indicating greater impairment or disability [21].

3) Quality of life (QOL) is defined as "An individual's perception of their position in life in the context of the culture in which they live and in relation to their goals, expectations, standards, and concerns" [22]. QOL is an important consideration for improving the symptom relief, care, and rehabilitation of patients [23].

4) Disabilities of the Arm, Shoulder and Hand (DASH)

is a self-administered questionnaire designed to measure physical function and symptoms in individuals with musculoskeletal disorders of the upper limb. It comprises 30 items that assess the impact of arm, shoulder, and hand impairments on daily activities, work, and recreational tasks. Each item is rated on a 5-point Likert scale, and the overall score is calculated and transformed to a scale ranging from 0 (no disability) to 100 (most severe disability). The DASH has been widely validated and is recognized for its reliability in clinical research and practice [24].

5) Global Rating of Change (GROC) Scale

is a single-item, patient-reported outcome measure used to capture an individual's overall perception of change in their condition following an intervention. Patients are asked to rate their change relative to baseline on a scale that typically ranges from –7 (a very great deal worse) to +7 (a very great deal better), with 0 indicating no change. This scale provides a comprehensive overview of treatment effectiveness by reflecting perceived improvements or deteriorations in pain, function, and quality of life [25].

## Study selection

Two independent researchers (YJK and SJJ) will oversee the study selection process while adhering to the outlined criteria (Table 1). After eliminating duplicate entries, YJK and SJJ will scrutinize the titles and abstracts of the retrieved studies to determine their relevance, and thoroughly evaluate the full texts of the selected studies for eligibility. Any disagreements regarding the study selection will be resolved through consultation with other researchers. Our approach to literature selection will be documented in accordance with the PRISMA guidelines (Fig 1) [26].

**Fig 1. PRISMA-compliant flow diagram depicting selection of studies.**

## Data extraction

Data extracted will encompass the primary author's name, publication year, country of origin, paper title, sample size, dropout count, participants' age and sex, intervention and comparison specifics, adverse effects of the intervention, research design, measurement tools employed

and independent, dependent, mediated, and control variables as well as subfactors pertinent to shoulder joint pain.

Specifically, for data collection on intervention outcomes, in order to comprehensively understand the durability of the effects of MET, data on pain, joint range of motion, and quality of life will be collected at time points such as 3 months, 6 months, or even 1 year after treatment.

For instance, upon confirming that a study used METs or similar interventions, we will focus on quantifying the degree of improvement observed in shoulder joint pain based on the primary outcome measure. Subsequent considerations will include variables such as shoulder mobility and functional activity. Furthermore, studies that compare techniques, such as the Mulligan or Kaltenborn mobilization techniques, with METs, will be categorized and comparatively analyzed to assess their efficacy.

The compiled data will be systematically recorded using Excel 2024 (Microsoft, Redmond, WA, USA) and shared with the researchers using Dropbox folders (Dropbox Inc., San Francisco, CA, USA). If the data are deemed insufficient or ambiguous, correspondence with the respective authors of the included studies will be initiated via email to request supplementary information.

## Quality assessment

Two independent researchers (YJK and SJJ) will evaluate the methodological quality of the included studies and the quality of evidence of each primary finding. Discrepancies will be resolved through consultation with other researchers. The methodological quality of the studies will be assessed using the Cochrane risk-of-bias tool [27]. Random sequence generation, allocation concealment, blinding of participants and personnel, blinding of outcome assessments, incomplete outcome data, selective reporting, and other biases will be assessed in each study. Each domain will be categorized as "low risk," "unclear," or "high risk." The evaluation results will be documented in an Excel 2024 spreadsheet and shared among the researchers using Dropbox.

The assessed outcomes will be comprehensively presented in a full review using Review-Manager version 5.3 (Cochrane, London, UK). The quality of the evidence will be depicted through a summary of the findings. The evaluation procedure will be shared and discussed among researchers.

## Data synthesis and analysis

ReviewManager version 5.3 and Excel 2024 will be used for data synthesis and analysis. Researchers will collaborate using Dropbox folders to share files. Descriptive analyses of participant demographics, interventions, and outcomes will be conducted for all included studies. Studies involving comparable interventions, comparisons, and outcomes will be quantitatively synthesized. Data will be analyzed in two phases: (1) data synthesis and analysis after the systematic review process and (2) categorization of studies with figures suitable for meta-analysis.

The first step is a systematic review to comprehensively organize and analyze studies that show significant effects of METs on shoulder pain among patients. Each study will be classified and coded to "author (year of publication)," "participants (patients)," "difference in shoulder pain before and after muscle energy technique," "research methods," "research procedures," and "research result." Second, the effects of METs on shoulder joint pain described in the selected papers will be systematized through discussions and reviews among the researchers.

To assess the correlation between shoulder pain and the MET score in each study, we will designate and code the analyzed items as follows: First, we will categorize and compare outcomes based on the techniques applied to patients with shoulder pain, while distinguishing MET from other techniques as variables. Second, in studies using only the MET, we will define

MET application and non-application groups as variables and compare the differences in outcomes. Third, in studies examining the application of the MET in patients with shoulder joint pain, we will identify correlation codes, review theoretical backgrounds, classify each variable into an analyzable framework, and synthesize sub-variables. Subsequently, we will analyze the overall publication bias, verify homogeneity, analyze the overall effect size, and investigate the correlation effect size among all factors related to the rehabilitation period. The size of the correlation effects will be analyzed using Fisher's z value [28] (0.1 = small effect size, 0.3 = medium effect size, and 0.5 = large effect size) by calculating the correlation coefficient with a 95% confidence interval. We will evaluate the heterogeneity among the studies using both the chi-square test and the I-squared statistic. I-squared values > 50% and > 75% will be considered indicative of substantial and high heterogeneity, respectively. A random-effects model will be applied when significant heterogeneity is detected (I-squared value > 75%), whereas a fixed-effects model will be employed when heterogeneity is not significant, or if the number of studies included in the meta-analysis is very small and inter-study variance estimates lack precision. [29] If heterogeneity is deemed too substantial for synthesis (I-squared value > 75%), a subgroup analysis will be conducted to elucidate the source of heterogeneity.

### Assessing the quality of the body of evidence

The quality of evidence will be evaluated using the Grading of Recommendations, Assessment, Development, and Evaluation (GRADE) [30] framework across five categories: risk of bias, imprecision, inconsistency, indirectness, and other factors such as publication bias.

### Subgroup analysis

If heterogeneity is significant (I-squared value > 75%) and the necessary data are available, we will perform a subgroup analysis to account for heterogeneity. We will also conduct a subgroup analysis for the following potential factors: age, sex, race, session duration, different levels and types of joint mobilization, and other factors that may affect the results.

### Sensitivity analysis

To assess the robustness of the meta-analysis results, sensitivity analyses will be conducted by excluding 1) studies with a high risk of bias, 2) studies with missing data, and 3) outliers.

### Assessment of reporting bias

If the analysis includes more than 10 trials, reporting biases, including publication bias, will be assessed using funnel plots and the trim-and-fill method. If asymmetry is observed in the funnel plots, indicating a potential reporting bias, further investigation will be conducted to identify the possible causes of the asymmetry.

### Ethics and dissemination

Ethical approval is unnecessary as individual patient data will not be included in this systematic review. The findings will be shared through publication of the manuscript in a peer-reviewed journal and/or presentation at a pertinent conference.

## Discussion

The primary treatment goals of rehabilitation therapy are pain relief and ROM restoration. Particularly in the shoulder joint, which demands a wide ROM and is predominantly utilized in daily activities, pain and limited ROM can significantly affect an individual's functional performance, making rehabilitation therapy a major focus in clinical practice [31]. Shoulder

pain in patients with shoulder disorders reduces shoulder ROM, thereby compromising QOL [32]. Therefore, establishing effective treatment interventions is crucial for shoulder joint pain rehabilitation. MET is broadly applicable to various musculoskeletal conditions, with relatively immediate and sustained effects and minimal patient discomfort [33,34]. Therefore, it is considered useful for reducing muscle tension, improving pain, and restoring joint ROM. However, to date, no systematic literature review has been conducted on the effects of the MET on shoulder joint pain and function. This study aims to analyze the effect size of the MET on shoulder joint pain and its contribution to shoulder joint mobility, function, and improvement in patients' quality of life. We believe that the results of this systematic review will provide clinical evidence for the use of the MET in treating restricted shoulder joint mobility. Furthermore, by proposing effective methods for shoulder rehabilitation, we hope to offer valuable and cost-effective alternatives for the healthcare sector. This systematic review and meta-analysis will have several limitations. First, significant heterogeneity is anticipated among the included RCTs. Variations in participant inclusion criteria, intervention protocols, assessment tools, and follow-up duration across studies may influence the overall interpretation of the findings. To address this, subgroup analyses using consistent criteria will be conducted to minimize heterogeneity. Second, there is a considerable risk of bias. Although the quality of the included studies will be assessed using the Cochrane RoB 2.0, the studies may not have been methodologically robust in terms of design or execution. Third, the use of diverse outcome assessment methods across the studies presents some challenges. Differences in follow-up periods and tools used to evaluate outcomes such as pain and functional recovery may complicate the aggregation of the results. To mitigate this issue, we will standardize the outcome measures or utilize statistical methods that allow for the integration of diverse outcomes. Fourth, the small sample sizes of some RCTs may lead to inadequate statistical power. Smaller studies are more prone to random errors, which could adversely affect the reliability of meta-analysis results. To compensate for this, wider confidence intervals will be used. Considering these limitations, caution is advised when interpreting the findings of this review, and larger-scale RCTs are required in future research to strengthen the evidence base.

## Supporting information

**S1 Appendix. Search terms and strategies.**
(DOCX)

**S1 Checklist. PRISMA-P Checklist.**
(PDF)

## Acknowledgments

The authors have no acknowledgments to declare.

## Author contributions

**Conceptualization:** Ye Ji Kim, Seojae Jeon.

**Data curation:** Seojae Jeon.

**Formal analysis:** Ye Ji Kim.

**Investigation:** Ye Ji Kim, Seojae Jeon, Junghan Lee.

**Methodology:** Seojae Jeon.

**Project administration:** Junghan Lee.

**Resources:** Ye Ji Kim, Seojae Jeon, Junghan Lee.

**Supervision:** Seojae Jeon, Junghan Lee.

**Validation:** Hyeonjun Woo, Won-Bae Ha, Junghan Lee.

**Visualization:** Seojae Jeon.

**Writing – original draft:** Ye Ji Kim.

**Writing – review & editing:** Ye Ji Kim, Seojae Jeon.

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
