## [Decision Letter · Decision Letter 0]

13 Aug 2024

PONE-D-24-23377A protocol for a systematic review and meta-analysis of the effect of muscle energy techniques on shoulder joint painPLOS ONE

Dear Dr. Jeon,

Thank you for submitting your manuscript to PLOS ONE. After careful consideration, we feel that it has merit but does not fully meet PLOS ONE’s publication criteria as it currently stands. Therefore, we invite you to submit a revised version of the manuscript that addresses the points raised during the review process.

The authors should enhance the methodology of their systematic review protocol by including specific search strategies for each database to be utilized, employing a valid risk of bias assessment tool, and detailing the methods for conducting the meta-analysis.

We look forward to receiving your revised manuscript.

Kind regards,

Mohammadreza Pourahmadi, PT, Ph.D., Postdoctoral Fellow

Academic Editor

PLOS ONE

Additional Editor Comments:

Reviewers' comments:

Reviewer's Responses to Questions

**Comments to the Author**

1. Does the manuscript provide a valid rationale for the proposed study, with clearly identified and justified research questions?

Reviewer #1: Yes

Reviewer #2: Partly

2. Is the protocol technically sound and planned in a manner that will lead to a meaningful outcome and allow testing the stated hypotheses?

Reviewer #1: Yes

Reviewer #2: Partly

3. Is the methodology feasible and described in sufficient detail to allow the work to be replicable?

Reviewer #1: Yes

Reviewer #2: Yes

4. Have the authors described where all data underlying the findings will be made available when the study is complete?

Reviewer #1: Yes

Reviewer #2: Yes

5. Is the manuscript presented in an intelligible fashion and written in standard English?

Reviewer #1: Yes

Reviewer #2: Yes

6. Review Comments to the Author

You may also provide optional suggestions and comments to authors that they might find helpful in planning their study.

Reviewer #1: The authors present a systematic review protocol to collect clinical evidence regarding the effectiveness of muscle

energy techniques on glenohumeral joint pain.

Some suggestions for the manuscript to be revised:

Introduction:

After the sentence, "Randomized controlled trials (RCTs) have been conducted on METs.", authors must include two references of RCTs published in the last 10 years at least.

Methods:

I suggest that the authors change this to: Studies selected for systematic review and meta-analysis will include randomized controlled clinical trials and quasi-randomized controlled trials, avoiding the inclusion of non-randomized and observational studies. In line 132, just leave "The systematic review and meta-analysis will include RCTs and quasi-randomized controlled trials."

On line 154, cite a reference that talks about ROM. In line 196, you must insert the reference to the Cochrane Rob2.0 tool.

Discussion:

The authors must introduce a paragraph with the possible limitations of the future review, such as high heterogeneity between RCTs, high risk of bias, different ways of evaluating outcomes in trials, such as follow-up time, instruments used and small number of patients in RCTs compromising the quality of the results.

Reviewer #2: 1. Please provide a search strategy for each database.

2. You mentioned that a variety of research types will be included, including: randomized controlled clinical trials, quasi-randomized controlled trials, and controlled (non-randomized) clinical trials. However, the inclusion of multiple studies will lead to increased heterogeneity. How to deal with it?

3. It is unclear whether the study follows the PICOS framework.

4. Cochrane Collaboration's risk-of-bias tool is for RCTs. How do non-randomized controlled trials assess quality? You didn't provide further details.

5.The protocol suggests utilizing funnel plots to evaluate publication bias; however, this technique may have its drawbacks. Incorporating other methods, such as the trim-and-fill method or cumulative meta-analysis, could yield more reliable insights regarding publication bias. A thorough evaluation of publication bias is essential to guarantee the dependability of the meta-analysis results.

6. Limitations of this study should be added to the discussion section.

7. PLOS authors have the option to publish the peer review history of their article (what does this mean? ). If published, this will include your full peer review and any attached files.

**Do you want your identity to be public for this peer review?** For information about this choice, including consent withdrawal, please see our Privacy Policy .

Reviewer #1: **Yes: ** Ricardo Ney Cobucci

Reviewer #2: No

---

## [Author Response · Author response to Decision Letter 1]

3 Oct 2024

Reviewer #1: The authors present a systematic review protocol to collect clinical evidence regarding the effectiveness of muscle energy techniques on glenohumeral joint pain.

Some suggestions for the manuscript to be revised:

Introduction:

After the sentence, "Randomized controlled trials (RCTs) have been conducted on METs.", authors must include two references of RCTs published in the last 10 years at least.

- I added references.

Methods:

I suggest that the authors change this to: Studies selected for systematic review and meta-analysis will include randomized controlled clinical trials and quasi-randomized controlled trials, avoiding the inclusion of non-randomized and observational studies. In line 132, just leave "The systematic review and meta-analysis will include RCTs and quasi-randomized controlled trials."

- I changed it.

On line 154, cite a reference that talks about ROM. In line 196, you must insert the reference to the Cochrane Rob2.0 tool.

- I added references.

Discussion:

The authors must introduce a paragraph with the possible limitations of the future review, such as high heterogeneity between RCTs, high risk of bias, different ways of evaluating outcomes in trials, such as follow-up time, instruments used and small number of patients in RCTs compromising the quality of the results.

- I added possible limitations on discussion part.

Reviewer #2: 1. Please provide a search strategy for each database.

- I added it with Supporting information (Appendix)

2. You mentioned that a variety of research types will be included, including: randomized controlled clinical trials, quasi-randomized controlled trials, and controlled (non-randomized) clinical trials. However, the inclusion of multiple studies will lead to increased heterogeneity. How to deal with it?

- I excluded controlled (non-randomized) clinical trials.

3. It is unclear whether the study follows the PICOS framework.

- I added it with table.

4. Cochrane Collaboration's risk-of-bias tool is for RCTs. How do non-randomized controlled trials assess quality? You didn't provide further details.

- I added it.

5.The protocol suggests utilizing funnel plots to evaluate publication bias; however, this technique may have its drawbacks. Incorporating other methods, such as the trim-and-fill method or cumulative meta-analysis, could yield more reliable insights regarding publication bias. A thorough evaluation of publication bias is essential to guarantee the dependability of the meta-analysis results.

- I added trim and fill.

6. Limitations of this study should be added to the discussion section.

- I added some limitations in discussion part.

7. PLOS authors have the option to publish the peer review history of their article (what does this mean?). If published, this will include your full peer review and any attached files.

---

## [Decision Letter · Decision Letter 1]

26 Oct 2024

PONE-D-24-23377R1A protocol for a systematic review and meta-analysis of the effect of muscle energy techniques on shoulder joint painPLOS ONE

Dear Dr. Jeon,

Thank you for submitting your manuscript to PLOS ONE. After careful consideration, we feel that it has merit but does not fully meet PLOS ONE’s publication criteria as it currently stands. Therefore, we invite you to submit a revised version of the manuscript that addresses the points raised during the review process.

After reviewing the revised manuscript, the reviewers still had several minor comments, particularly concerning typographical errors and some aspects of the study's methods.

We look forward to receiving your revised manuscript.

Kind regards,

Mohammadreza Pourahmadi, PT, Ph.D., Postdoctoral Fellow

Academic Editor

PLOS ONE

Journal Requirements:

Reviewers' comments:

Reviewer's Responses to Questions

**Comments to the Author**

1. Does the manuscript provide a valid rationale for the proposed study, with clearly identified and justified research questions?

Reviewer #1: Yes

Reviewer #2: Yes

2. Is the protocol technically sound and planned in a manner that will lead to a meaningful outcome and allow testing the stated hypotheses?

Reviewer #1: Yes

Reviewer #2: Yes

3. Is the methodology feasible and described in sufficient detail to allow the work to be replicable?

Reviewer #1: Yes

Reviewer #2: Yes

4. Have the authors described where all data underlying the findings will be made available when the study is complete?

Reviewer #1: Yes

Reviewer #2: Yes

5. Is the manuscript presented in an intelligible fashion and written in standard English?

Reviewer #1: No

Reviewer #2: Yes

6. Review Comments to the Author

You may also provide optional suggestions and comments to authors that they might find helpful in planning their study.

Reviewer #1: The authors have met most of the recommendations, but there are still flaws in the revised manuscript.

"Studies selected for systematic review and meta-analysis will include randomized controlled clinical trials, quasi-randomized controlled trials, and controlled (nonrandomized) clinical trials" in lines 61 and 62 needs to be corrected, as well as the text in Table 1 Research method Randomized controlled clinical trials, quasi-randomized controlled trials, nonrandomized controlled clinical trials (except for qualitative research and case studies).

Improve the new text of the inclusion and exclusion criteria in lines 134 and 135.

Finally, there are word errors throughout the manuscript and the quality of the scientific writing needs to be improved, with the suggestion that the entire text be reviewed by a native speaker, or a professional language editing service.

Reviewer #2: The paper conducts research on the impact of muscle energy techniques on shoulder joint pain. The research objective is clear and the methods are detailed. The authors have also made good revisions. However, there are still some issues that need to be addressed, which are as follows:

1. The "point" in the subtitle "Effect of muscle energy techniques on shoulder point pain: a protocol" may be a typo of "joint".

2. You said you no longer included non-randomized clinical trials, but you didn't delete "controlled (non-randomized) clinical trials" from the methods section of your ABSTRACT.

3. Appropriate simple descriptions of the expected research results can be added in the abstract, such as what level of evidence support is expected to be provided for the application of MET techniques in the treatment of shoulder joint pain through the research.

4. Now it is October 2024, and it is not appropriate for "comprehensively search the following databases from their inception up to April 2024", please modify it.

7. PLOS authors have the option to publish the peer review history of their article (what does this mean? ). If published, this will include your full peer review and any attached files.

**Do you want your identity to be public for this peer review?** For information about this choice, including consent withdrawal, please see our Privacy Policy .

Reviewer #1: **Yes: ** Ricardo Ney Cobucci

Reviewer #2: No

---

## [Author Response · Author response to Decision Letter 2]

11 Nov 2024

Response to Reviewers

Dear Editor and Reviewers,

We sincerely appreciate the valuable comments and suggestions provided by the reviewers, which have significantly helped us to improve the quality of our manuscript. We have addressed each of the reviewers’ comments in detail below. Furthermore, we would like to inform you that the entire manuscript has been professionally edited by Editage, a reputable language editing service, to ensure clarity and accuracy in English language usage. We have also received an official certificate from Editage to verify this process, which we are prepared to submit if required.

Response to Reviewer #1

Comment 1: “Studies selected for systematic review and meta-analysis will include randomized controlled clinical trials, quasi-randomized controlled trials, and controlled (nonrandomized) clinical trials” (lines 61 and 62) needs to be corrected, as well as the text in Table 1 (Research method: Randomized controlled clinical trials, quasi-randomized controlled trials, nonrandomized controlled clinical trials, except for qualitative research and case studies).

Response: We have made the requested corrections to the text in both lines 61-62 and Table 1. In the revised manuscript, we have clarified that only randomized controlled trials will be included, as per the latest inclusion criteria. Non-randomized controlled trials have been excluded from our review criteria to ensure methodological rigor.

Comment 2: Improve the new text of the inclusion and exclusion criteria in lines 134 and 135.

Response: We have refined the wording of the inclusion and exclusion criteria in lines 134 and 135 to enhance clarity and precision. The updated criteria explicitly describe the type of studies included and excluded to align with our study objectives and methodology.

Comment 3: There are word errors throughout the manuscript, and the quality of the scientific writing needs to be improved. It is suggested that the entire text be reviewed by a native speaker or a professional language editing service.

Response: We acknowledge this suggestion and have had the entire manuscript professionally edited by Editage. This editing process has corrected any language errors and improved the overall quality of scientific writing. As mentioned, we have received a certificate from Editage confirming the completion of this professional editing service.

Response to Reviewer #2

Comment 1: The "point" in the subtitle "Effect of muscle energy techniques on shoulder point pain: a protocol" may be a typo of "joint."

Response: We apologize for this oversight. We have corrected the subtitle to "Effect of muscle energy techniques on shoulder joint pain: a protocol."

Comment 2: You said you no longer included non-randomized clinical trials, but you didn't delete "controlled (non-randomized) clinical trials" from the methods section of your ABSTRACT.

Response: Thank you for pointing this out. We have revised the methods section of the abstract to exclude any mention of non-randomized clinical trials, aligning it with the revised inclusion criteria.

Comment 3: Appropriate simple descriptions of the expected research results can be added in the abstract, such as what level of evidence support is expected to be provided for the application of MET techniques in the treatment of shoulder joint pain through the research.

Response: We appreciate this suggestion. We have added a brief description in the abstract regarding the anticipated impact of our study, specifically mentioning the potential contribution of evidence supporting the use of MET techniques in treating shoulder joint pain.

Comment 4: Now it is October 2024, and it is not appropriate for "comprehensively search the following databases from their inception up to April 2024". Please modify it.

Response: This has been updated to reflect the current time frame, with the revised text stating that the search will be conducted up to October 2024.

We hope that the revisions and explanations provided here satisfactorily address all the comments and suggestions from the reviewers. We appreciate the opportunity to improve our manuscript and thank the reviewers again for their constructive feedback.

Best regards,

Ye Ji Kim

---

## [Decision Letter · Decision Letter 2]

15 Jan 2025

PONE-D-24-23377R2A protocol for a systematic review and meta-analysis of the effect of muscle energy techniques on shoulder joint painPLOS ONE

Dear Dr. Jeon,

Thank you for submitting your manuscript to PLOS ONE. After careful consideration, we feel that it has merit but does not fully meet PLOS ONE’s publication criteria as it currently stands. Therefore, we invite you to submit a revised version of the manuscript that addresses the points raised during the review process.

Dear Authors,

Thank you for your efforts in revising the manuscript in accordance with the reviewers’ comments. Reviewer 2 has highlighted a point that requires further improvement in the manuscript. Please revise the manuscript accordingly and resubmit it.

We look forward to receiving your revised manuscript.

Kind regards,

Seyed Hamed Mousavi

Academic Editor

PLOS ONE

Journal Requirements:

Reviewers' comments:

Reviewer's Responses to Questions

**Comments to the Author**

1. Does the manuscript provide a valid rationale for the proposed study, with clearly identified and justified research questions?

Reviewer #1: Yes

Reviewer #2: Yes

Reviewer #3: Yes

2. Is the protocol technically sound and planned in a manner that will lead to a meaningful outcome and allow testing the stated hypotheses?

Reviewer #1: Yes

Reviewer #2: Yes

Reviewer #3: Yes

3. Is the methodology feasible and described in sufficient detail to allow the work to be replicable?

Reviewer #1: Yes

Reviewer #2: Yes

Reviewer #3: Yes

4. Have the authors described where all data underlying the findings will be made available when the study is complete?

Reviewer #1: Yes

Reviewer #2: Yes

Reviewer #3: Yes

5. Is the manuscript presented in an intelligible fashion and written in standard English?

Reviewer #1: Yes

Reviewer #2: Yes

Reviewer #3: Yes

6. Review Comments to the Author

You may also provide optional suggestions and comments to authors that they might find helpful in planning their study.

Reviewer #1: The authors met most of the reviewers' recommendations and the manuscript can be accepted. Congratulations

Reviewer #2: The article has been revised very well. Thank you for your efforts. There is still one point that needs to be improved.

Supplement of Long-term Follow-up and Safety Assessment: The current research protocol mainly focuses on the short-term efficacy of MET on shoulder joint pain and function, but lacks consideration of its long-term effects. A plan for long-term follow-up of patients should be added. For example, the indicators such as pain, range of motion of the joint, and quality of life should be evaluated at 3 months, 6 months, or even 1 year after treatment to comprehensively understand the durability of the efficacy of MET. At the same time, the safety assessment during the MET treatment process is not involved in the article. Observation indicators and methods should be supplemented to record possible adverse reactions (such as muscle strains, excessive joint movement, etc.) to ensure the safety of this treatment method in clinical application.

Reviewer #3: Following the review of the author’s reply to the concerns that were raised previously, I recommend approval of the article for publication. Thank you

7. PLOS authors have the option to publish the peer review history of their article (what does this mean? ). If published, this will include your full peer review and any attached files.

**Do you want your identity to be public for this peer review?** For information about this choice, including consent withdrawal, please see our Privacy Policy .

Reviewer #1: **Yes: ** Ricardo Ney Cobucci

Reviewer #2: No

Reviewer #3: **Yes: ** Collins Ogbeivor

---

## [Author Response · Author response to Decision Letter 3]

25 Feb 2025

Thank you for reviewing my paper. I have revised it based on your valuable revision suggestions. Please check it. Thank you.

---

## [Decision Letter · Decision Letter 3]

3 Mar 2025

A protocol for a systematic review and meta-analysis of the effect of muscle energy techniques on shoulder joint pain

PONE-D-24-23377R3

Dear Dr. Jeon,

We’re pleased to inform you that your manuscript has been judged scientifically suitable for publication and will be formally accepted for publication once it meets all outstanding technical requirements.

Kind regards,

Seyed Hamed Mousavi

Academic Editor

PLOS ONE

Additional Editor Comments (optional):

Reviewers' comments:

Reviewer's Responses to Questions

**Comments to the Author**

1. Does the manuscript provide a valid rationale for the proposed study, with clearly identified and justified research questions?

Reviewer #2: Yes

2. Is the protocol technically sound and planned in a manner that will lead to a meaningful outcome and allow testing the stated hypotheses?

Reviewer #2: Yes

3. Is the methodology feasible and described in sufficient detail to allow the work to be replicable?

Reviewer #2: Yes

4. Have the authors described where all data underlying the findings will be made available when the study is complete?

Reviewer #2: Yes

5. Is the manuscript presented in an intelligible fashion and written in standard English?

Reviewer #2: Yes

6. Review Comments to the Author

You may also provide optional suggestions and comments to authors that they might find helpful in planning their study.

Reviewer #2: I greatly appreciate the author's efforts in revising the article. The article has been revised very well, so I recommend it for publication.

7. PLOS authors have the option to publish the peer review history of their article (what does this mean? ). If published, this will include your full peer review and any attached files.

**Do you want your identity to be public for this peer review?** For information about this choice, including consent withdrawal, please see our Privacy Policy .

Reviewer #2: No

---

## [Editor Report · Acceptance letter]

PONE-D-24-23377R3

PLOS ONE

Dear Dr. Jeon,

I'm pleased to inform you that your manuscript has been deemed suitable for publication in PLOS ONE. Congratulations! Your manuscript is now being handed over to our production team.

Kind regards,

on behalf of

Dr. Seyed Hamed Mousavi

Academic Editor

PLOS ONE